# Comparative Analysis of Bivalve and Sea Urchin Genetics and Development: Investigating the Dichotomy in Bilateria

**DOI:** 10.3390/ijms242417163

**Published:** 2023-12-05

**Authors:** Anatoliy Drozdov, Egor Lebedev, Leonid Adonin

**Affiliations:** 1Zhirmunsky National Scientific Center of Marine Biology, Far Eastern Branch of the Russian Academy of Sciences, 690041 Vladivostok, Russia; 2Institute of Environmental and Agricultural Biology (X-BIO), Tyumen State University, 625003 Tyumen, Russia; mainsalta@gmail.com; 3Institute of Biomedical Chemistry, 119121 Moscow, Russia

**Keywords:** protostomia, deuterostomia, mollusks, sea urchin, genome, transposable elements (TE), non-coding RNA (ncRNA), transcription, chromatin landscape

## Abstract

This comprehensive review presents a comparative analysis of early embryogenesis in Protostomia and Deuterostomia, the first of which exhibit a mosaic pattern of development, where cells are fated deterministically, while Deuterostomia display a regulatory pattern of development, where the fate of cells is indeterminate. Despite these fundamental differences, there are common transcriptional mechanisms that underline their evolutionary linkages, particularly in the field of functional genomics. By elucidating both conserved and unique regulatory strategies, this review provides essential insights into the comparative embryology and developmental dynamics of these groups. The objective of this review is to clarify the shared and distinctive characteristics of transcriptional regulatory mechanisms. This will contribute to the extensive areas of functional genomics, evolutionary biology and developmental biology, and possibly lay the foundation for future research and discussion on this seminal topic.

## 1. Introduction

Embryogenesis is a complex and tightly regulated series of events in which a fertilized egg develops into a complex organism. The architectural blueprints for the mechanisms that control this process are intricately encoded in the genome. The regulation of transcription plays a crucial role in orchestrating the precise patterns of gene expression that drive embryonic development. Understanding the similarities and differences in the mechanisms that regulate transcription during early and later embryogenesis in different animal groups can provide valuable insights into the evolutionary history and developmental processes of these organisms.

Metazoans are divided into Protostomia and Deuterostomia based on their embryonic development [1,2,3] (Figure 1). The divergence between these groups occurred during the early Cambrian period, over 538.8 million years ago [4,5,6]. Protostomia, which include Arthropoda, Mollusca and Annelida, exhibit mosaic development, typically with spiral cleavage and a determinate fate of embryonic cells [1]. In contrast, Deuterostomia, including Chordata, Echinodermata and Hemichordata, follow a regulatory development with radial cleavage and an indeterminate fate of cells [7,8,9,10]. Members of these branches of the phylogenetic tree show notable disparities in their embryonic development, characterized by divergent types of gastrulation and outcomes for the blastopore [9,11]. Nevertheless, both Protostomia and Deuterostomia also share fundamental traits in their embryonic development, such as the constitution of germ layers and the orientation of body axes.

The main works in the study of the genome structure and the various mechanisms of its regulation in the early embryogenesis processes in Protostomia and Deuterostomia are based on data obtained from classical model organisms belonging to these two groups: Drosophila, nematodes, mice and rats. However, our focus is on representatives of bivalve mollusks (Bivalvia, Mollusca) and sea urchins (Echinoidea, Echinodermata). These two groups of animals are classic objects of embryology, belonging to the two main branches of the animal kingdom (Metazoa). Mollusks are typical primary animals with a spiral cleavage (Spiralia, Protostomia), whereas echinoderms are typical secondary animals with a radial cleavage (Deuterostomia).

This review aims to comprehensively compare and contrast the mechanisms involved in transcriptional regulation during early embryogenesis in Protostomia and Deuterostomia. The representative taxa Echinoidea and Bivalvia are examined and analyzed as illustrative examples to accomplish this objective.

While the role of evolutionary gene regulatory elements has been studied quite well in bivalves [15,16] and especially well in sea urchins [17,18], we focused primarily on the regulation of non-coding RNAs. We will also examine the role of chromatin remodeling and certain transcription factors in shaping the gene expression landscape during embryonic development. Through an integrated literature review and innovative research, this article adds to the current understanding and creates a platform for further discussion on the topic.

## 2. Comparative Embryogenesis of Bivalves and Sea Urchins: Insights and Distinctive Characteristics

Bivalve mollusks and echinoderms are both higher triploblastic multicellular animals possessing a well-developed coelom, a derived feature of the third germ layer, the mesoderm [19]. These animals belong to distinct phyla of higher multicellular organisms. Bivalve mollusks demonstrate mosaic development, while echinoderms exhibit regulative development. These two animal groups exhibit both forms of development in a classical and typical manner, devoid of any notable evolutionary complexity or modifications [20,21].

Mollusks are Protostomia, while echinoderms are Deuterostomia animals. Their fundamental pattern of zygote cleavage (spiral holoblastic cleavage in mollusks and radial holoblastic cleavage in echinoderms) and the method of mesoderm formation (teloblastic in mollusks and enterocoele in echinoderms) distinguish them. The organization of the nervous system, the type of larvae, and the position of the mouth and anus during embryonic and post-embryonic development also differentiate Protostomia and Deuterostomia. The larvae of mollusks exhibit a trochophore morphology, whereas echinoderms display bipinnaria larvae. Grobben [22] first defined these two branches [19]. Furthermore, the deuterostome’s skeleton is of mesodermal origin, whereas the protostome’s skeleton is of ectodermal origin. The skin of deuterostomes comprises two layers, including an ectodermal epithelium and mesodermal connective tissue.

In sea urchins, as in all echinoderms, the first two cleavages occur in the meridional plane, resulting in the formation of four identical blastomeres. The third division is equatorial. The resulting eight blastomeres are arranged strictly radially in two rows, one above the other. Cleavage leads to the formation of a ciliated blastula, and the embryo begins to move inside the envelope. Shortly after, the larvae emerge from the envelopes and start an actively swimming lifestyle due to the beating of cilia. Then, cells from the vegetative part of the blastula undergo immigration, and the vegetative wall of the blastula bulges into the blastocoel. The process of gastrulation occurs through invagination and ends with the formation of a narrow cylindrical archenteron, from which cells of the secondary mesenchyme migrate. The blastopore in echinoderms is located at the posterior end of the larva. The cells of the primary and secondary mesenchyme participate in the formation of the mesoderm, but the major portion of the mesoderm arises through the invagination of the archenteron. A vesicle, the precursor of the coelom, buds off from the upper end of the primary gut, and it then divides into two and positions itself laterally to the archenteron. The archenteron bends towards the wall of the gastrula, which is now referred to as the ventral side. Subsequently, the blastopore shifts to this side, and the larva becomes bilaterally symmetric, with a concave ventral side and a convex dorsal side. The larva is referred to as a bipinnaria, which subsequently develops into the planktotrophic larva known as an echinopluteus in sea urchins. The pluteus mainly feeds on microorganisms such as bacteria and phytoplankton, but it is also capable of assimilating dissolved organic matter found in seawater.

Sea urchin embryos have served as a classical model for studying regulative development. It is possible to obtain two, three, or four fully formed larvae at the pluteus stage by dividing the first two or four blastomeres through mechanical shaking, decalcified or hypertonic seawater [23,24,25]. Additionally, monozygotic twins in sea urchins can be produced through centrifugation [26,27]. The production of monozygotic twins in sea urchins is straightforward by exposing zygotes to centrifugation within 6 min of fertilization. Introducing cytochalasin B to fertilized eggs in seawater vastly increases the possibility of twin formation through centrifugation between fertilization and the initial cleavage furrow, as documented by Drozdov and Svyatogor [27].

Typical Protostomia, such as bivalves, exhibit spiral and determinate development, referred to as “mosaic” development [28]. 

In Spiralia (Mollusca and Annelida), blastomeres are situated in close proximity during early embryonic development. This cleavage process results in the formation of a stereoblastula, with limited expression or absence of a blastocoel in various mollusk and annelid species [29,30]. These organisms exhibit a distinct form of invagination termed dense ingrowth, in which macromeres extend along the animal-vegetative axis of the embryo, ultimately filling the blastocoel. Following this, cells of the inner layer produce the endodermal rudiment, also referred to as the primary gut or archenteron, through the process of delamination [31].

The embryo is viewed as a mosaic of preformed primordia that are already present in the unfertilized egg. However, there is evidence that Spiralian embryos possess certain regulatory potencies. In earthworms (Eisenia), it has been possible to induce twinning deformities by mechanically altering the shape of the eggs [32]. The number of deformities increases when cytochalasin B is added to the environment [33]. In the sea hare (Aplysia), twinning deformities have been induced by centrifuging fertilized eggs [34]. When subjected to centrifugation for 5 min at 2000 g, bivalve mollusk embryos (*C. gigas* and *Mactra chinensis*) and the chiton Ischnochiton hakodadensis have shown cases of polyembryony, and the occurrence of polyembryonic cases increases with centrifugation in the presence of cytochalasin B also [27].

At the close of the 19th century, the mosaic theory of embryonic development surfaced, positing that each blastomere bears information about predetermined body parts in distinct zones of the zygote. This theory has been substantiated by over a century of experimental embryology trials on embryos of Protostomia (such as mollusks and polychaetes) [28].

In Deuterostomia, which include echinoderms, lancelets, fish, and amphibians, individual totipotent blastomeres are capable of developing into fully normal larvae, with certain qualifications. Initially, a blastula is formed, which then progresses through the gastrula stage before eventually transforming into a dwarf pluteus, as described by Driesch [23] and Wilson [28]. Zoja [35] obtained similar results in the hydrozoan Clytia. The process of egg pre-organization puzzled scientists for quite some time until the concept of positional information was introduced in the latter half of the 20th century. The concept of positional information [36,37,38] serves as a universal model to explain the formation of morphogenesis patterns during embryonic development and regeneration [39,40]. This hypothesis suggests that morphogenetic determinants, namely morphogens-ribonucleoprotein (RNP) particles, present in the cytoplasm of the egg, zygote, and blastomeres, direct the differentiation of embryos [21,41,42,43]. Morphogens are formed during the entire embryonic process, with their initial localization established upon fertilization, and play a critical role throughout organismal development. This process could lead to cytoplasmic segregation [44,45,46].

One approach toward analyzing the mechanisms of morphogenesis is the study of polyembryony, including the use of centrifugation [27,47,48]. We have shown that the development of twins in sea urchin zygotes subjected to centrifugation until the formation of the first two blastomeres can be explained by changes in the localization of morphogenetic determinants leading to the division of the embryo at the blastula stage into two halves, each of which undergoes gastrulation and subsequently develops into a dwarf pluteus [27]. In some cases, the blastula forms not one but two additional axes, resulting in the appearance of three archenterons. However, these gastrulae do not develop into larvae.

It has been shown that unfertilized sea urchin eggs contain globular actin and a network of short microfilaments in the cortex [42]. After fertilization, these elements associate, resulting in the formation of a robust cortical cytoskeleton that plays a crucial role in morphogenesis prior to the first cleavage.

A robust network of actin microfilaments is present in the cortex of unfertilized bivalve eggs [49]. The strong cortical cytoskeleton in mollusks is formed during oogenesis and serves to localize morphogenetic determinants, which undergo minimal reorganization after fertilization, ensuring determinant or so-called “mosaic” development [21].

## 3. Post-Genomic Era of Bivalve and Sea Urchins Studying

In comparative genomics, a key priority is revealing the structural and functional features that are innate to the genomes of Protostomia and Deuterostomia taxa. The existence of reference genomes for model organisms has facilitated the description of their distinct features, such as gene arrangement, regulatory motifs, and repetitive elements. This enables us to elucidate the complex signaling pathways and processes responsible for early development, adaptation, regeneration and overall organismal complexity.

The study of genomes has greatly transformed our comprehension of the genetic basis of life. In the field of mollusks, 200 genomes have been sequenced, with 77 of these belonging to the Bivalvia taxa. One of the inaugural molluscan genomes to be sequenced was that of the Pacific oyster, *Crassostrea gigas*, which was accomplished in 2012 [50]. Fourteen sea urchin genomes have been sequenced, including that of the purple sea urchin, *Strongylocentrotus purpuratus* [51,52,53], which was one of the first to have its complete genome sequenced and assembled. The sequencing of the *S. purpuratus* genome in 2006 marked the start of a new era in sea urchin research [51,52,54,55,56].

### 3.1. Genome Size and Complexity

Most genome size studies have focused on vertebrates, leaving invertebrates relatively understudied [57]. Sequencing of certain bivalve mollusks has uncovered remarkably high levels of genome heterozygosity [58]. For instance, *C. gigas* has been found to have a heterozygosity rate of 1.3% [50], *Ruditapes philippinarum* exhibits a rate of 2.0% [59], and the quagga mussel *Dreissena rostriformis* displays a rate of 2.4% [60]. This places these organisms among the animal species with the greatest genetic diversity [61,62]. The genome size of bivalves varies from 0.65 pg/N to 5.4 pg/N across 108 species, while for sea urchins, it ranges from 0.54 pg/N to 1.3 pg/N across 21 species (Gregory, T.R. (2023) Animal Genome Size Data: www.genomesize.com). Genomic assembly data for the first sequenced genome, as well as the largest and smallest genomes, of three different species of sea urchins and bivalves are available in Table 1.

### 3.2. Gene Organization and Density

The genome of *S. purpuratus* is about 900 Mb in size and contains about 33,500 protein-coding genes [56]. Some of these genes were previously considered as innovations in vertebrates or were known only in Protostomia [63,64,65]. Analysis of the genome of another sea urchin species, *Paracentrotus lividus*, and its comparison with the genomes of *S. purpuratus* and *Lytechinus variegatus* [66] revealed that the gene order in sea urchin genomes evolves differently compared to vertebrates. A recent analysis [41] of genomes, epigenomes, and transcriptomes in two sea urchin species, *Heliocidaris erythrogramma*, which undergoes direct development from gastrula to adult without feeding, [67] and *Heliocidaris tuberculata*, which undergoes indirect development with a feeding larva [68], revealed significant differences in the expression profiles of early developmental genes between these closely related species. These differences are primarily explained by adaptive changes in the sequences of presumed regulatory elements and the regulation of their chromatin state, which may result in a reduction or delay in the transcription of numerous genes during zygotic development. 

The rate of interchromosomal rearrangements appears to be very low, while the rate of change in the local gene order within chromosomes seems to be much faster. This observed difference may be attributed to a relaxation of functional and regulatory constraints on gene order compared to vertebrates [41].

The current genome of *C. gigas* contains approximately 31,000 protein-coding genes. Comparative analysis with the genome of the pearl oyster *Pinctada fucata* [69] demonstrates that genes shared among bivalves encompass a greater number of genes that potentially relate to the extracellular matrix, environmental responses, and the immune system compared to those observed in gastropod mollusks and other Protostomia. A study on chromosomal macrosynteny between the bivalves *Mactra veneriformis* and *Patinopecten yessoensis* revealed that, similar to sea urchins, bivalves have a relatively low frequency of interchromosomal rearrangements and a high frequency of intrachromosomal rearrangements [70].

## 4. Repetitive Elements and Non-Coding RNAs

Repetitive elements (REs), including transposable elements (TEs, transposon), can serve as a major source of genomic changes in eukaryotic organisms, contributing to the generation of new genetic material and promoting species diversity and innovation [71,72]. TEs are repetitive DNA sequences capable of moving within the genome [73].

Currently, it is common to classify TEs based on their structural characteristics and mode of replication [74,75]. There are two main classes of TEs: Class I TEs, also known as retrotransposons, which transpose through an RNA intermediary; and Class II TEs, also known as DNA transposons, which use a conservative cut-and-paste mechanism where the donor element is excised and then reinserted at a different location in the genome [76].

Bivalve mollusks, in particular, have been found to possess a notably high diversity of transposons compared to other mollusk species. For instance, the average observed transposable element content in the genomes of 27 bivalve species is reported to be 38.97%. However, Pectinidae exhibit a relatively lower proportion of TEs, at around 20% [77]. Compared to the genomes of cephalopods and gastropods, there is a higher prevalence of LINE retrotransposons, specifically RTE-X and CR1-Zenon. The presence of SINEs does not follow a consistent pattern; however, V elements can constitute up to 4.3% of the genome in the mussel *Bathymodiolus platifrons*. LTR types, such as Bel/Pao, DIRS, and Ngaro, are predominantly found in bivalves and gastropods. In contrast, the Gypsy and Copia elements are present across all mollusk classes. LTR Steamer-like elements show widespread transfer throughout bivalves’ evolutionary history, occurring in vertebrates, sea urchins, and corals [78].

The types of DNA transposons, namely Kolobok, Zator, and Academ, are mainly found in bivalves, whereas the Zisrupton, Novosib, and Merlin superfamilies are predominantly restricted to the analyzed cephalopods [77].

On the other hand, sea urchins show a relatively stable average TE composition. For example, in the species of sea urchin *S. purpuratus*, the composition of TEs, including unclassified ones, is 31.5% of the genome, and in *L. variegatus* it is 35.9% [79] (Figure 2). While the majority of the repetitive fraction of vertebrate genomes typically consists of retrotransposons [80], DNA transposons are dominant in both sea urchins and bivalves.

TE activity needs to be tightly regulated, especially during embryogenesis, to avoid negative effects on the developing organism. However, some TEs conversely form regulatory networks during embryonic development; for example, it has been shown that transposons in the genome of *S. purpuratus* significantly change their expression during gastrulation [81].

REs are involved in a significant proportion of the transcription of non-coding RNAs (ncRNAs) [82,83,84,85,86].

ncRNAs are a diverse group of RNA molecules that do not encode proteins but play essential roles in various cellular processes. They have emerged as key regulators of gene expression and have been implicated in numerous biological functions [87,88,89,90,91]. ncRNAs can be classified according to their size and mechanism of action [92,93]. Small ncRNAs (sncRNAs), such as microRNAs (miRNAs) and small interfering RNAs (siRNAs), are typically 20 to 30 nucleotides in length and function by binding to target messenger RNAs (mRNAs), leading to their degradation or translational repression [94]. Another class of sncRNAs are PIWI-interacting RNAs (piRNAs), which are involved in transposon silencing in the germ lines [95,96].

Long non-coding RNAs (lncRNAs) and long intergenic non-coding RNA (lincRNA), on the other hand, are more than 200 nucleotides in length and have diverse roles, including regulating the chromatin structure, modulating transcription and interacting with proteins [97,98]. Other classes of ncRNAs include small nucleolar RNAs (snoRNAs), which direct chemical modifications of other RNAs, and circular RNAs (circRNAs), which have been implicated in gene regulation and as potential biomarkers [99]. The analysis and identification of non-coding RNAs have significantly revolutionized our understanding of gene regulation [100].

Research has shown that different genomic loci, which are responsible for the production of primary piRNAs, exhibit selective activity during specific developmental phases in the Pacific oyster (*C. gigas*, Bivalvia). In contrast, the same loci that generate piRNAs are active exclusively in the germ line and somatic cells in great pond snails (*L. stagnalis*, Gastropoda). The four piRNA clusters identified displayed considerable diversity, both in terms of the overall structure and TE composition. Additionally, there was a noticeable accumulation of young Gypsy elements in the piRNA clusters, indicating a strong preference for the insertion of Gypsy elements, potentially due to their activity within the genome of *C. gigas* [101].

piRNAs are involved in embryonic development not only by silencing transposons but also by interacting with mRNAs [102,103]. Thus, for *S. purpuratus,* it has been shown that Seawi, a homologue of the PIWI, has been observed to regulate proliferation of primordial germ cells, apparently by altering Vasa protein expression during embryogenesis [104].

The role of miRNAs has also been demonstrated in bivalve developmental processes. Specifically, the expression levels of miRNA biogenesis genes within C. gigas show remarkable upregulation during the initial stages of oyster development, including the period from two-cell formation to rotary movement. In some cases, a high expression level was also observed in subsequent developmental stages up until the D-shaped larvae phase. In contrast, adult oysters primarily display low basal expression of these genes [105]. The participation of microRNAs in larval development has also been shown for the pearl oyster, *Pinctada fucata* [106], and the razor clam, *Sinonovacula constricta* [107].

Post-transcriptional regulation by miRNAs is important for early embryonic developmental mechanisms in the sea urchin *S. purpuratus* [108,109] and *Hemicentrotus pulcherrimus* [110], including through the regulation of skeletogenesis [111,112] and the modulation of β-catenin in the canonical Wnt signaling pathway [113].

There is increasing evidence for the lncRNA-mediated regulation of embryonic development [114]. For example, a paper by Yu and colleagues investigated the dynamics of expression of lincRNAs in the *C. gigas* at 35 time points of different developmental stages (egg to juvenile). A total of 10,685 lincRNAs were identified, of which 809 showed specific expression. Analysis of the lincRNA-mRNA co-expression network showed that five lincRNAs can function in larval development [115].

Sequencing of the sea urchin *P. lividus* genome [66] revealed intriguing features of intrachromosomal gene order mixing. In addition, more than 5000 lncRNAs were annotated in transcriptomes for 17 embryonic stages (from fertilized egg to pluteus stage), and 5 adult tissues, most of which showed stage-specific expression.

## 5. Comparative Genomic Analyses

The Doublesex-mab3-related transcription factor (Dmrt) gene family is a class of crucial transcription factors. Dmrt family genes can participate in various physiological developmental processes, especially in sex determination. A comparative analysis identified that Dmrt1-like is specific to the bivalve genome and is involved in sex differentiation [116]. Indeed, a more recent analysis of Dmrt genes in 11 echinoderms, including 3 sea urchins, did not reveal a Dmrt1-like gene. Echinoderms were characterized by Dmrt2-like, Dmrt3-like, Dmrt4/5-like, Dsx-like and novel Dmrt class, specific for sea stars [117].

Forkhead (Fox) family transcription factors have been identified in many metazoans and play important roles in a variety of biological processes. They have a regulatory function in the process of gene expression and influence development, differentiation, metabolism, and immunity [118]. In the *C. gigas* genome, the expression of 26 Fox genes has been detected [119]. Two of these genes, Cg06159 and Cg24546, are specifically expressed in the ovaries. The highest expression of these two Fox genes is observed in the oocytes, and their expression levels gradually decrease during embryonic development until they become undetectable by the late D-stage (Cg24546) or juvenile stage (Cg06159). In sea urchins, FoxQ2, which is a homolog of Cg06159, plays a crucial role in ectoderm patterning in embryos of *S. purpuratus* and *H. pulcherrimus* [55,120]. On the other hand, Cg24546 is closely related to FoxN2. In sea urchins, FoxN2/3 is essential for several steps in the formation of the larval skeleton [121].

Hox genes are a group of genes that encode transcription factors that are important in the regulation of a variety of developmental processes in the organism. They play an important role in pattern formation of the body axis and in the differentiation of cells, tissues and organs [122]. 

A Hox cluster containing 11 Hox genes was found in the genome of *Lottia gigantea* [123] and further in two genomes of scallop *Chlamys farreri* [124] and Patinopecten yessoensis [125,126]. Whole-genome analysis of the Hox gene family of the mussel *M. coruscus* revealed that 11 Hox genes are evenly distributed within a single chromosome [127]. Candidate orthologues of ten Hox genes, except for Hox7, as in *C. gigas* [50], and three ParaHox sister complex genes were identified from the transcriptome of *Dreissena rostriformis* [128]. Three ParaHox genes have also been identified from the oysters *Pinctada fucata* [69] and *C. gigas* [50,129]. 

The Hox cluster in sea urchins is by far the most thoroughly studied [66,122]. The *S. purpuratus* Hox cluster consists of 11 genes (Hox4 has not been identified) [54]. *S. purpuratus* also contains homologues of the three ParaHox genes [130]. In the irregularly shaped urchin *Peronella japonica* [131] and the sea urchin *L. variegatus*, the same rearranged Hox cluster as in *S. purpuratus* was detected [66].

## 6. Conclusions

Therefore, even though Protostomia and Deuterostomia, such as bivalves and sea urchins, originated more than half a billion years ago, they persistently exhibit distinct developmental characteristics that set them apart from each other. For example, protostomes exhibit mosaic development with spiral cleavage and a determinate cellular fate, while deuterostomes undergo radial cleavage and develop indeterminate cells. In addition, members of these taxa also exhibit disparities in various other aspects, such as the fate of the blastopore, coelom formation, nervous system development, and regenerative abilities. 

Sequencing genomes at a high resolution level enabled the assessment of patterns and features in composition of genes and repetitive elements. Various regulatory strategies that are crucial for embryogenesis have been examined, including gene-mediated regulation, the role of repetitive elements, and different classes of noncoding RNAs that are differentially expressed at various developmental stages. A high rate of local gene order change within chromosomes has been observed in members of these taxa, relative to vertebrates. 

To comprehend the adaptations that account for the remarkable range of mechanisms controlling embryogenesis, it may be necessary to scrutinize the adaptive alterations of regulatory element sequences, such as noncoding RNAs.

However, despite extensive research, our understanding of the multifaceted layers of complexity and regulatory networks that impact transcriptional mechanisms remains incomplete. To gain deeper insight into these divergent developments, a more thorough analysis of genomic data is required. This will enable us to elucidate the overarching principles governing these mechanisms and advance our knowledge in this field.

## Figures and Tables

**Figure 1 ijms-24-17163-f001:**
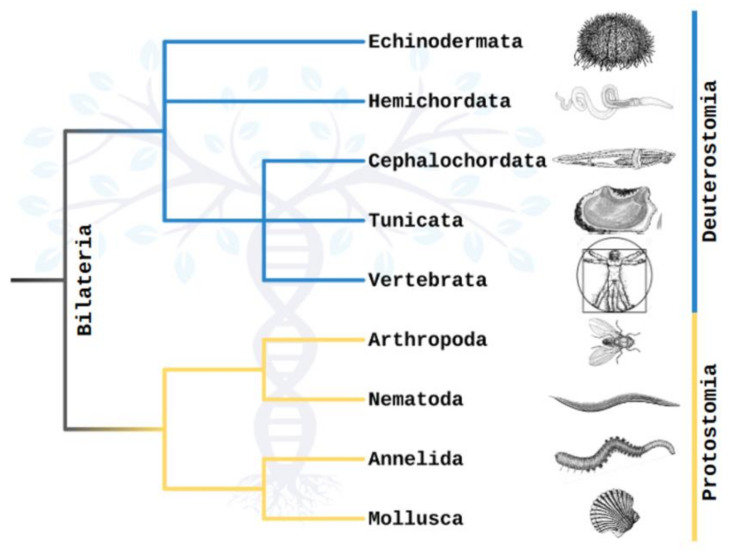
Truncated phylogenetic tree representing major taxa of the Protostomia and Deuterostomia, derived from combined analyses of molecular and morphological data [12,13,14]. Branch lengths do not correspond with time.

**Figure 2 ijms-24-17163-f002:**
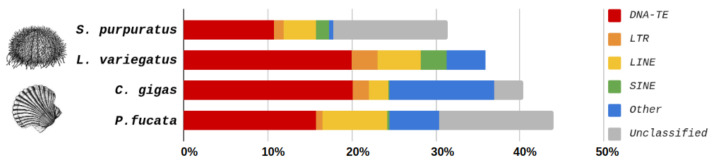
The percent content of transposable elements (TE) in genomes of the sea urchins and bivalves. TE classes are marked with a color: DNA transposons—red; LTR TE—ochre; nonLTR: LINE—yellow; SINE—green; other TE—blue; unclassified repeats—gray.

**Table 1 ijms-24-17163-t001:** Key features of annotated bivalves and sea urchin genomes.

	Name	Assembly	GenBank	Genome Size	Assembly Level
Sea urchins	*S. purpuratus*	Spur_5.0	GCA_000002235.4	921.8 Mb	Scaffold
	*E. tribuloides*	Etri_1.0	GCA_001188425.1	2.2 Gb	Scaffold
	*H. pulcherrimus*	HpulGenome_v1	GCA_003118195.1	568.9 Mb	Scaffold
Bivalves	*C. gigas*	cgigas_uk_roslin_v1	GCA_902806645.1	647.9 Mb	Chromosome
	*S. cumingii*	ASM2855479v1	GCA_028554795.1	3.4 Gb	Chromosome
	*L. rhynchaena*	LuRhyn_1.0	GCA_008271625.1	543.9 Mb	Contig

## Data Availability

Not applicable.

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
