# Peer review of "Comparative Analysis of Bivalve and Sea Urchin Genetics and Development: Investigating the Dichotomy in Bilateria"

_ijms, 2023, doi:10.3390/ijms242417163_

Round 1

Reviewer 1 Report

Comments and Suggestions for Authors

The goal of this review was to compare and contrast early development in protostomes and deuterostomes, with an emphasis on recent genomic and molecular advances.  Specific topics addressed included genomic architecture and non-coding RNAs.

Strengths

The section on repetitive elements and non-coding RNAs (section 3.1.3) was particularly informative and discussed recent findings and compared multiple protostome and deuterostome species. The paragraphs on the role of miRNAs was well done.

Major Comments:

1.     The paragraph that outlines the goals of the review (lines 60-67), mentions that the review will compare and contrast mechanisms of transcriptional regulation, the maternal-zygotic transition and the midblastula transition.  But it wasn’t clear to me that any of these three topics was directly addressed in the review.  For example, the terms maternal-zygotic transition and midblastula transition are only mentioned once in the review—in that paragraph.  

2.     Section 3.1.1 and 3.1.2 describe many features of the genomes. How do these genomic differences relate to differences in development?  What is the broader functional importance of these differences?

3.     The Conclusion mentions that more research needs to be done on the evolution of regulatory elements and how regulatory networks impact transcription (lines 364-368).  A lot of work has been done enhancers and transcription factor regulatory networks in sea urchins, and that work was not included in this review.  This review seems to focus on regulatory RNAs—is that what is meant by regulatory elements?  Considering the comment from #1 above and this comment on the Conclusion, I am surprised this review did not touch on the evolution gene regulatory element (enhancers and transcription factors).  If that literature is outside the goals/scope of this review, it would be good to make that clear in the introduction.

Author Response

Dear reviewer, 

We greatly appreciate the time and effort you have dedicated to reviewing our manuscript, and we would like to thank you for your constructive comments which have brought an important issue to our attention.

  • The paragraph that outlines the goals of the review (lines 60-67), mentions that the review will compare and contrast mechanisms of transcriptional regulation, the maternal-zygotic transition and the midblastula transition.  But it wasn’t clear to me that any of these three topics was directly addressed in the review.  For example, the terms maternal-zygotic transition and midblastula transition are only mentioned once in the review—in that paragraph.  

Thank you for your valuable feedback regarding the paragraph outlining the goals of our review (lines 60-67). We completely agree with your observation that the topics of transcriptional regulation, the maternal-zygotic transition, and the midblastula transition were not directly addressed in the review as explicitly as stated in that paragraph. We apologize for any confusion caused by the introductory mention of these terms without thorough subsequent exploration.

In light of your comments, we have revised that particular section to align with the specific aspects of the review that are extensively discussed. We have removed the misleading reference to "the maternal-zygotic transition and the midblastula transition" and, instead, focused on the more prominent themes that emerged during our analysis of the literature.

  • Section 3.1.1 and 3.1.2 describe many features of the genomes. How do these genomic differences relate to differences in development?  What is the broader functional importance of these differences?

In response to your remark, we have restructured and combined sections 3.1.1 and 3.1.2 into a singular, cohesive section that better explicates the functional implications of genomic differences in relation to development. We have carefully considered your inquiry about the broader functional importance of the described genomic differences.

Upon further review and analysis, we find that there is no straightforward correlation between genome size and specific developmental features. However, our data support a distinct correlation between genome size and the abundance and diversity of transposable elements within the genomes in question. This finding is intriguing as it suggests a potential indirect relationship between genome structure and developmental processes. 

Transposable elements, as you are aware, can serve as regulatory elements, influencing gene expression patterns during both embryonic and post-embryonic development. Their ability to induce mutations, alter gene expressions, and even restructure the genome can have profound implications on the developmental trajectories of organisms. We posit that this role of transposable elements could serve as a mediator between the genome and developmental characteristics. Hence, while the genome size itself does not directly correlate with developmental nuances, the mediated effects through the transposon-related regulatory changes could be of significant functional importance.

  • The Conclusion mentions that more research needs to be done on the evolution of regulatory elements and how regulatory networks impact transcription (lines 364-368).  A lot of work has been done enhancers and transcription factor regulatory networks in sea urchins, and that work was not included in this review.  This review seems to focus on regulatory RNAs—is that what is meant by regulatory elements?  Considering the comment from #1 above and this comment on the Conclusion, I am surprised this review did not touch on the evolution gene regulatory element (enhancers and transcription factors).  If that literature is outside the goals/scope of this review, it would be good to make that clear in the introduction.

We would like to clarify that while our review briefly mentions various regulatory elements, our primary focus was indeed on regulatory RNAs and their role within the intricate landscape of embryonic gene regulation. This focus was chosen due to the emerging significance of regulatory RNAs in recent literature and the relatively less explored dimension they offer in the context of embryogenesis.

In response to your comments, we have revisited the introduction of our manuscript to more explicitly state the scope and goals of our review.

Sincerely yours,

Drozdov Anatoliy, Lebedev Egor and Adonin Leonid.

Reviewer 2 Report

Comments and Suggestions for Authors

The Dichotomy in Bilateria: Unraveling the Genetic and 2 Developmental Differences Between Protostomia and 3 Deuterostomia

Drozdov et al

This manuscript entail a review style manuscript and is aimed and ‘unraveling’ the genetic and developmental differences between Protostomia and Deuterostomia. This is a rather broad topic, that would provide sufficient information to cover a whole book, and this would still be insufficient. Indeed, the authors do not compare Deuterostomia with Protostomia; instead they have (fortunately) limited themselves  to sea urchins (representing Deuterostomia) and bivalve molluscs (representing Protostomia). My advice is o change the title accordingly, as it stand now it is misleading.

Figure 2 is a textbook image and redundant. It also has no relationship with the text and the main topic of the review.

In line 142, when Deuterostomia are considered there is a mentioning of plutei larvae. Clearly here a subset of deuterostomia is considered, amphibia for instance do not form plutei. Please make clear (in the title) that the manuscript is comparing sea urchins with bivalves, not Protostomia with Deuterostomia.

Lines 95-108: there is a rather detailed explanation of gastrulation in echinoderms, but a similar description of gastrulation in bivalve molluscs is lacking. This is however necessary if the authors want to make a comparison.

Ine 150 ‘Morphogens are formed during oogenesis’ . This is dependent on the definition of a morphogen. The general definition in embryology is a secreted factor (signalling factor) that diffused through tissue and can influence the behavior of responding cells as a result of the concentration. There are not formed during oogenesis, but during the entire embryonic process.

Table 1 and Figure 3 please indicate which are sea urchins and which are bivalves

Line 162-165. It is not clear what the function of this part of the text is in relation to the rest.

Comments on the Quality of English Language

sufficient

Author Response

Dear reviewer, 

We greatly appreciate the time and effort you have dedicated to reviewing our manuscript, and we would like to thank you for your constructive comments which have brought an important issue to our attention.

  • This manuscript entails a review style manuscript and is aimed and ‘unraveling’ the genetic and developmental differences between Protostomia and Deuterostomia. This is a rather broad topic, that would provide sufficient information to cover a whole book, and this would still be insufficient. Indeed, the authors do not compare Deuterostomia with Protostomia; instead they have (fortunately) limited themselves to sea urchins (representing Deuterostomia) and bivalve molluscs (representing Protostomia). My advice is to change the title accordingly, as it stand now it is misleading. 

Upon reflection, we concede that the original title of our manuscript may indeed seem overambitious and could give the impression of a scope far beyond what has been addressed in the content of our review. Your critique aptly highlights that the breadth of genetic and developmental differences between all Protostomia and Deuterostomia is a subject extensive enough to fill volumes and that our focused examination on sea urchins and bivalve molluscs provides a mere snapshot of this immensely complex field. 

Thus, in accordance with your advice and to better reflect the specific focus of our comparative analysis, we have revised the title of our manuscript. We believe that the new title accurately encapsulates the narrowed scope of our study and sets clear expectations for the reader.

We acknowledge that a less encompassing title will more accurately represent the content of our article and negate any potential misunderstandings regarding the breadth of our review. We are committed to ensuring that our research is presented as transparently and accurately as possible, and we are hopeful that this change will improve the overall reception and clarity of our work.

  • Figure 2 is a textbook image and redundant. It also has no relationship with the text and the main topic of the review.

We agree that Figure 2 is indeed a textbook image and may be considered redundant for illustrating the main topic of our review. We understand that it does not contribute significantly to the overall understanding of our text, as it lacks a direct relationship with the main focus of our revuew.

We acknowledge that utilizing this particular image may not be necessary, and we are open to removing it from the manuscript. Your observation has prompted us to reconsider its inclusion, and we will ensure that any revisions made align with the aim of enhancing the clarity and relevance of the content.

  • In line 142, when Deuterostomia are considered there is a mentioning of plutei larvae. Clearly here a subset of deuterostomia is considered, amphibia for instance do not form plutei. Please make clear (in the title) that the manuscript is comparing sea urchins with bivalves, not Protostomia with Deuterostomia. 

We would like to assure you that our objective indeed centers around comparing the embryogenic processes of bivalve molluscs and sea urchins. After considering your comment, we have revised the title to accurately reflect the emphasis of our study. The updated title now clearly specifies the comparison between bivalve molluscs and sea urchins.“2. Comparative Embryogenesis of Bivalves and Sea Urchins: Insights and Distinctive Characteristics”.

  • Lines 95-108: there is a rather detailed explanation of gastrulation in echinoderms, but a similar description of gastrulation in bivalve molluscs is lacking. This is however necessary if the authors want to make a comparison.

We would like to express our gratitude for bringing this to our attention, as it allowed us to rectify this omission promptly. In response, we have added a detailed explanation of gastrulation in bivalve molluscs to ensure a more balanced and thorough comparison. The additional content has been highlighted in yellow for ease of reference.

“In Spiralia (Mollusca and Annelida), blastomeres are situated in close proximity during early embryonic development. This cleavage process results in the formation of a stereoblastula, with limited expression or absence of a blastocoel in various mollusk and annelid species (Drozdov, 1983; Drozdov et al., 1983). These organisms exhibit a distinct form of invagination termed dense ingrowth, in which macromeres extend along the animal-vegetative axis of the embryo, ultimately filling the blastocoel. Following this, cells of the inner layer produce the endodermal rudiment, also referred to as the primary gut or archenteron, through the process of delamination (Malakhov, Medvedeva, 1991)”.

  • Line 150 ‘Morphogens are formed during oogenesis’. This is dependent on the definition of a morphogen. The general definition in embryology is a secreted factor (signaling factor) that diffuses through tissue and can influence the behavior of responding cells as a result of the concentration. There are not formed during oogenesis, but during the entire embryonic process.

In light of your observation, we have revised the pertinent sentence to more accurately reflect the correct embryological concept of morphogens. The revised statement now reads, "Morphogens are formed during the entire embryonic process, with their initial localization established upon fertilization, and play a critical role throughout organismal development." All changes in the text have been highlighted in yellow for ease of reference.

  • Table 1 and Figure 3 please indicate which are sea urchins and which are bivalves. 

Thank you for your valuable feedback. We have taken your suggestion into consideration and made the necessary revisions to Table 1 and Figure 3 (Figure 2 now). In Figure 2, we have included schematic illustrations of a sea urchin and a bivalve next to their respective Latin names. This will clearly indicate which organisms are sea urchins and which are bivalves. 

Additionally, we have divided Table 1 into two distinct sections: one for mollusks and the other for sea urchins. This separation will provide a clear distinction between the two groups and facilitate easier identification of the respective organisms.

We appreciate your attention to detail and believe that these modifications will enhance the clarity and comprehensibility of the table and figure. Thank you for bringing this to our attention, and we are grateful for your input.

  • Line 162-165. It is not clear what the function of this part of the text is in relation to the rest. 

Thank you for your comment regarding the unclear function of a particular section of the text in relation to the rest. We acknowledge and agree that the paragraph in question does not contribute cohesively to the main body of the text. Its inclusion in the article was an oversight on our part during the final formatting of the manuscript. We have removed this paragraph in the revised version of the manuscript.

Sincerely yours,

Drozdov Anatoliy, Lebedev Egor and Adonin Leonid.
